# The Scramble for Religion and Secularism in Pre-Colonial Africa

Benson Ohihon Igboin

Department of Religion and African Culture, Adekunle Ajasin University, Akungba-Akoko 342111, Nigeria; benson.igboin@aaua.edu.ng

**Abstract:** The debate on the existence of religion in Africa is far from over; it reverberates in new dimensions but asking the same old questions in newer ways. The same argument is being extended to secularism. This article takes a critical look at the concepts, religion and secularism in sub-Saharan pre-colonial Africa, raising the recurring question still maintained by the West whether there was 'religion' in Africa at the turn of colonialism. It argues that where no religion exists, the notion of secularism as understood by the West, cannot also exist since the latter is not just the 'opposite' of the former, but in actual fact, streams from it. However, since the position of 'non-religion' in Africa could not be sustained by the West, the question of how and why 'secularism' was not also 'discovered' in pre-colonial Africa spontaneously arises. Not denying the widespread diversities of religious beliefs in pre-colonial Africa, this paper argues for the existence of religion, and presence and praxis of religio-secularity, which is non-atheistic in nature that also foregrounds the practice of 'secularity' in post-colonial Africa.

**Keywords:** secularism; secularity; secularisation; religion; pre-colonial Africa; ATR

## 1. Introduction

The search for 'religion' in/of sub-Saharan Africa is not yet over after decades of European colonial and missionary authorities claimed there was no religion in Africa and also later claimed there was religion. This search continues to reverberate in contemporary scholarship and relationship; it comes in ever dynamic resonations, either with the aim to historically ground its 'discovery' or contest its deeply pervasive presence both in the past and present. The earliest writings on sub-Saharan Africa, and specifically on religion in Africa, were obviously not carried out by scholars; they were essentially travelogues and missionary and colonial collections that have been found to be largely prejudicial and inaccurate. Those accounts were tailored towards representing Africa as objects to be exploited, converted and colonised. Such mindset definitely had a targeted public in the West, and as object, Africa was presented as 'dark continent', without the idea of God, religion, culture and history, and in dire need of civilisation, the ostensible aim of colonisation (Wijsen 2020; Adogame 2022). Western colonialism is not a finished product, it is a tortuous project, endlessly designed to rout out African collective heritage and development. For instance, several decades after independence, Francophone countries still pay colonial tax to France (Koutonin 2014). History of French and Belgian colonialist horrendous destruction of lives and property in their colonies still stirs pains in post-colonial reminiscence (Koutonin 2014). This is not exonerating other colonial authorities' dehumanisation of Africans and the constant tension of neo-coloniality.

The 'discovery' of 'religion' and philosophy in Africa by Western scholars took place in late colonial era when there was a shift from Africa as Object to Africa as Subject. Even at that, there was also a sort of colonisation of memory and politics of epistemology and methodology since the study of religion in Africa was dominated by the Europeans (Adogame 2022). The discovery and study of religion in Africa were both subjected to

European methodological rudiments, and therefore, unable to avoid misconceptions and misrepresentations of African 'religion'.

However, it appears that the Europeans did not simultaneously discover secularism as they did religion. According to Adrian Hastings (1985, p. 173), "The secularism of African tradition is a dimension still inadequately explored." Hastings notes that a lot of arguments have gone into 'religion' in Africa to the neglect of secularism. Why is secularism a late comer in the scheme of academic debate in Africa even after the issue of religion had largely been engaged? Why did the Europeans not analyse secularism until it became a reaction to African religious incurability? (Platvoet and van Rinsum 2003, 2008). In any case, the talks on the possibility of secularism in Africa have been countered by African deep religious incurability thesis. Both strands have continued to have impact on how religion is studied in Africa. This article is therefore poised to examine the trajectories of 'religion' and 'secularism' in pre-colonial Africa at the turn of colonialism.

Historical and analytical methods are utilised in the paper. Historical sources are garnered from accounts of the earliest European anthropologists, travel accounts, missionary accounts and surveys. These have been subjected to analysis with the broad objective of ascertaining the presence of religiosity and secularism in pre-colonial Africa. Throughout this paper, pre-colonial Africa will be used to refer to sub-Saharan Africa. In addition, I recognise the heterogeneity of African religion and culture, although it is more convenient to use them in the singular (see Ray 2000; Igboin 2021). From the analysis of these accounts and arguments that have been generated therefrom, I will argue that despite the permeability of religion in African societies, there were elements of what could be referred to as religio-secularity. In religio-secularity, there is no watertight institutional separation of social and political institutions from religious institution. Where social and political institutions existed, they functioned in accordance with rules and regulations, which were essentially not independent of the religious order. As Wariboko (2022a, p. 50) correctly puts it in my view, "there is no readily available distinction between religious and secular spheres" in different African societies. As we will exemplify later with some societies, African deities did not just provide spiritual satisfaction to the adherents, they also provided fun, social, emotional and personal security in the community (Pobee 1977, p. 1). Religio-secularity thesis is significant for understanding most contemporary African countries where religion still plays critical roles in public policies in spite of the claim that they are practising (constitutional) secularity. Christianity and Islam have ossified the presence of religion in the public space, taking over from the influence of indigenous religion. In other words, even though African indigenous religion does not seem to play significant role in post-colonial government in African countries, African Christians and Muslims have effectively taken over the public roles African indigenous religion played in the pre-colonial Africa.

In my opinion, religio-secularity differs from secularism in that the latter either aligns with atheism, religious nones, philosophical and ideological disenchantment with religion or binary split between the profane and the sacred, this-worldly and other-worldly. Such disenchantment and binary split provide the ground for secularity, a practical move towards separating the state from religion. Secularisation also differs from religio-secularity because it talks about the decline or privatisation of religion. As it will be made clear later, secularism, secularity and secularisation as they are understood in modern Western society might not have been the case in pre-colonial Africa. The possibility of religio-secularity rests on the concept of multiple secularities. Multiple secularities as a project recognise that secularism or secularity can be conceived differently from the Western conceptualisations on the one hand, and can be explored beyond the Western clime, which has dominated the study at present (Kleine and Wohlrab-Sahr 2020).

This article is divided into five sections. Section 1 is the introduction. Section 2 raises and analyses the problems associated with religion in Africa from a historical point of view. It argues that the earliest Europeans were hesitant to acknowledge the presence and existence of religion in Africa. Even when they later acknowledged to have found some

form of belief and belief system, its name—African Traditional Religion(s)—added to the complications of studying it. This is because the Western concept of religion was exclusively employed to interrogate the African belief. Section 3 stems from the second, arguing that since African belief system was hardly recognised as religion, it has implications on the presence of secularism in pre-colonial Africa. The reason is that both religion and secularism were regarded as too sophisticated for Africans to contemplate. However, the debate on religious incurability in Africa brought to the fore the issue of secularism—understood as the possibility of atheists and religious nones in pre-colonial Africa. Although there are no empirical evidence to substantiate the claims—religious incurability and secularism, Section 4 provides some ethnic examples to argue for the presence and praxis of religio-secularity. Section 5 concludes that the pre-colonial religio-secularity notion is still much present in contemporary Africa despite the claim of secularity.

## 2. The Problematic of 'Religion' in Africa

The first thing many scholars have tried to do is to (dis)entangle themselves with the elusive attempt at defining religion. Although the futility that has resulted from such attempt has not precluded the identification of religion where it exists and its meaning in context, it raises other concern bordering on acclaimed scientific methodology and episte-mological approaches in studying a phenomenon that largely defies definition, especially in other contexts. The whole idea of the quality of any religion cannot be fully empirically grasped. However, as Eric Sharpe (1983) has long asserted, our inability to define religion does not preclude our recognition of it when we come across one. Apart from that, the politics of identification and meaningfulness of religion has shaped, for a long time, what constituted religion in Africa, with the obvious implication that religion is a concept with superior meaning beyond the mental grasps of Africans. However, one can define religion as the awareness, recognition, consciousness of, and belief in, the existence of a supreme being to whom worship is due.

The problem of how to perfectly situate religion, "an emic term of European prove-nance and a modern Western concept" in Africa arose early enough in colonial contact with Africa (Kleine and Wohlrab-Sahr 2020, p. 6). For instance, early Western missionaries, anthropologists and colonialists did not find any signs or elements of 'religion' in Africa, because for them, religion was a sophisticated, philosophical and ideological concept and practice that credulous Africans were incapable of conceptualising and comprehending, let alone possessing. They thought that Africans had no concept of God and no religious worship because to have belief in God would automatically lead to the act and experience of worship of God (Mbiti 1969; Idowu 1973; Landau 2015; Grillo et al. 2019). However, Chidester (1996) gives a conceptual analysis of the genealogy of 'religion' and 'religions' as thought by European travellers, missionaries, settlers and colonial government and how the Western Christian concept of religion was used as the basis of comparison with African religious traditions. Rather than orthopraxis that is more visible in pre-colonial Africa, Chidester argues that Christian orthodoxy was the yardstick the Europeans used to measure the existence of religion in pre-colonial Africa, and that such conceptual position is still maintained in post-colonial Africa. Chidester (2013) argues that apart from the 'otherness' that characterised religious studies in the 19th century, there is the need to produce a counter-history of religion advanced and established by the Europeans. He calcifies that a counter-production of knowledge of religion and religions is critical to the understanding of indigenous African religion because it was race rather than theology that undergirded European study of African indigenous religion.

It will therefore be interesting to determine why and when the European explorers, missionaries and colonialists found elements of belief system in Africa. With regard to why, Jordan Fenton (2022, p. 7) points out that early European ethnographers and anthro-pologists tremendously "assisted in the colonization of Africa during colonial times" by their research outputs. Platvoet (1996) notes that the accounts of Victorian travellers includ-ing Sir Samuel Baker, Richard Burton as well as Christian missionaries such as Thomas

Bowen and David Livingstone were largely inaccurate but serve as "information bank" (Adogame 2022, p. 4) for historical reconstruction. Livingstone's account, for example, facilitated more explorations with the overarching aim of establishing Christianity, commerce and civilisation (Livingstone 2014; Kaunda 2018). They wrote in strong denigrating terms to obliterate the reality of religion of/in Africa in order to make case for colonialism and mission. Accordingly, Landau (2015, p. 190) provides a graph of the transition of recognition thus:

They had no 'beliefs' in a spiritual domain, and certainly no 'belief system' involving a spiritual domain. It was only later . . . when there arose the need for a non-political but map-able 'system,' an institution useful for ruling people, that imperially sponsored research 'discovered' belief-systems. That was tribal religion or 'African Traditional Religion,' or 'ATR,' its purported vocabulary being just those terms and dispositions that had been moved into Christianity (but beheld in their supposedly pre-Christian sense(s)) (Landau 2015, p. 190).

With regard to when, it is important to underscore the point that much of pre-colonial Africa history was distorted by Europeans aimed at setting intellectual barrier between us and our past traditions. In obstructing African past traditions, colonial history was projected and magnified above African reality. These traditions which included governance, religion, culture and so forth stemmed from before Arab and European slave trades, from the 16th to the 19th centuries when European imperialism and colonialism officially started (Ekeh 2007). Africanus (1896)[1] works were foundational sources to the European invasion of Africa. Upon evaluation, Africanus' works have been found to reflect Arab's interest in Africa, and provided justification for invasion, especially of Songhai Empire, a flourishing empire invaded in 1591. Leo Frobenius (1913) quoted an article in a German newspaper in 1891 to the effect that Africans generally lacked the idea of God until the Mohammadan invasion. Further to the 19th century, many states in sub-Saharan Africa would be invaded. Uthman dan Fodio's accounts on the Hausa followed the earlier trajectories, justifying invasion of already established dynasties. Fodio's jihad is even more unjustifiable, having recognised that the Hausa people had accepted Islam, but organised themselves politically in line with their African reality. Fodio wanted an ultra-conservative state ideology tilted towards theocracy, which he succeeded in establishing until the British invasion in 1903 (Ekeh 2007).

In the late 18th and early 19th centuries, European travellers and explorers wrote about Africa in the same form their Muslim predecessors had carried out in many instances, claiming Africa had no belief in God. Their accounts were amplified by armchair ethnographers and evolutionary anthropologists who described Africa as dark, primitive, heathen, pre-modern and so on (Platvoet 1996; Idowu 1973). This was the first phase of the study of 'religion' in Africa. The second phase began in the late 19th century when there was decline of evolutionary and the rise of social anthropology. Although still armed with Western methodological and historiographical mindset, such scholars as John Middleton and Victor Turner emerged. During this period, African cosmological system was studied with colonialist interpretation. The third phase, which started in the 1950s onward marked the collaboration of Europeans and Africans in researching African history, religion and philosophy. Adogame (2022, p. 6) aptly notes that "the word *religion* is a latecomer to the scholarly discourse about Africa" (emphasis in the text).

What was 'religion' before it became 'religion'? is an intriguing question. Religion historically did not start out as a concept, but as a practice either private or public by a people in a given context. Religion started as soon as a people were conscious of the existence of the supernatural, which they wanted to establish a relationship with. For those who have investigated the etymological and evolutionary trajectories of the term religion (although this is not our focus in this paper), what is indubitable is that 'religion' as a concept was not an all-time one, but the concept evolved from human activities guided by some form of beliefs, traditions, customs or cultures. Even though the West will lay claim to the concept today, it is clear that religion as both an activity and concept was a process in

time. Following Brent Nongbri (2015), it sounds not only ludicrous, but also dispossessing to think that older traditions than Judaism would be regarded as non-religious, except they come under the rubrics of Western conceptualisation of the term. Following Wilfred (Smith [1963] 1991), religion ceased to retain its human feeling or relationship the moment it began to be conceptualised. It thus suggests that as it stands in contemporary society, human activities called religion can be separated from what is conceptually called religion, and both strands can stand aloof, and yet complete in themselves. Clearly, as I want to understand Smith, this is the thrust and goal of secularism or secularisation. Reasoned this way, Smith would have ousted non-Western religious activities or perceived the enactors of the religious activities as incapable of conceptualising religion. Of course, the reification of religion spontaneously links it with secularism (Nongbri 2015). However, the link between religion and secularism was not intricate because both terms were used to distinguish the state of the clergy: those in monastery and others out the monastery. The concepts did not assume strict dichotomy between sacred and profane or pure and impure.

In the late 1950s when religion was used to refer to African belief system, it was used in descriptive sense—African Traditional Religion. However, even the nomenclature—ATR— is in itself problematic for it is not a name that constitutes the belief and belief system of the Africans; it is a convenient description that still subtracts the beliefs from being regarded as religion similar to other world religions. It is intriguing because it is either rendered in singular—African Traditional Religion— or in plural—African Traditional Religions (Olabimtan 2003). For instance, Christianity, Islam, Judaism, Zoroastrianism, Hinduism, Sikhism, Jainism, Buddhism, Shinto, Taoism or Confucianism despite the Western influences they have had almost neatly capture the essences of these religions. Their names are just one word, not a phrase, as African Traditional Religion(s). Of all the world religions, only Africa's will have the word religion as an appellation as though all others are not religions. In other words, it would not be incongruous to have Jewish Traditional Religion (JTR) rather than Judaism or Christianity, Chinese Traditional Religion (CTR) rather than Taoism or Confucianism, and so forth.

Describing the African beliefs as African Traditional Religion controversially by Geoffrey Parrinder (Shaw 1990) was perceived as an honour rather than a smear, because it seemed then as though a functional descriptive moniker had been invented to countenance what Africans practised as religion. The ideas of 'discovery' and 'inventing are pungent in comprehending ATR (Mudimbe 1988). Perhaps now that we can talk about Christianities or Islams in the plural sense with the attendant rejections and criticisms, African Traditional Religion(s) was not conceived in positive terms contrary to Aderibigbe's (2022) contention. Of course, other world religions can talk about sects and denominations to describe the differences and diversities in their beliefs within their religions, and still retain their names in the singular, African Traditional Religion(s) did not have that honour; its heterogeneity and diversities were lumped up into a politically correct phrase—ATR. In any case, whilst scholars such as Parrinder (1949, 1954, 1969), Idowu (1973), Magesa (1997), Aderibigbe and Falola (2022) amongst others prefer to use religion in the singular sense, others such as Mbiti (1969), Platvoet (1996), Ray (2000), Adogame (2022) amongst other use it in the plural. It must be noted that Mbiti (1991) had used it in the singular sense. The fundamental argument for the two schools of thought is uniformity of, and similarity in, African religious system. However, the point is Africa is heterogenous (see Igboin 2021).

Whilst some Europeans scholars have argued that African Traditional Religion (ATR) is not to be regarded as world religion because it lacks a founder and scriptures such as Christianity or Buddhism, the question of the founder of Hinduism still looms large (Magesa 1997; Grillo et al. 2019, p. 18). These European scholars such as Engelke posit that it is only religion qua religion that can be classified as world religion. Engelke (2015) notes that the term religion was exclusively used to refer to Christianity until recently when it became a generic word for other religions. He also argues that the status of being a world religion is a precursor to contemplate secularism, which is a form of civilisation. For him, it is only advanced religion that can be categorised as world religion, and ATR is yet to attain that

level of development. The implication of this is that Africa cannot contemplate secularism as it lacks a world religion. For instance, the influential philosopher of religion, Professor John Hick, who rationally articulated the possibilities of religious pluralism supposedly based on global religions or world religions, did not have any recourse to ATR(s) in the tree of religions. Whilst the religions of the Asians were discussed and critiqued, Hick, in his philosophical disquisition of religious pluralism did not make reference to ATR. However, many accepted his arguments as universal, and standard to ambiguate religious pluralism in religious studies and philosophy departments and discourses. This has raised critical questions within the circle of African philosophy of religion as to why Hick deliberately or inadvertently omitted ATR, or whether he agreed with the school of thought that Africa did not have what he could consider to be religion (Igboin 2014, 2019a).

It must however be conceded that whilst other religions developed their myths and stories and codified or canonised them as scriptures, Africans still pass much of theirs orally. The 'traditional' inserted in the name of the African religion(s) carries the import of fixity, a belief cast in primitivity and antiquity rather than in dynamic and progressive tone, a notion that religious intersectionality has now seriously challenged. For this reason, many African scholars are abandoning 'traditional' for 'indigenous' articulating that the latter carries more nuanced and positive connotations to suggest continuities and discontinuities. I will argue that this tack of decolonisation is still cosmetic and peripheral. A radically fitting act of decolonisation should be a complete re-naming of the belief systems rather than fight over 'traditional' or 'indigenous', and also constitute and institute a broader project to collate and codify the myths and stories of their diverse beliefs as the Yoruba Ifa corpus and the Aruosa Temple in Benin City, Nigeria have carried out, and broadly in tandem with the plethora of scriptures in Hinduism. As if sympathetic to this cause Grillo et al. (2019, p. 19) contend:

Viewing religion as a dynamic and coherent system organized to achieve such ends can make it clear that African indigenous religions are indeed comparable to the so-called world religions. It also allows us to identify common ways of knowing and being without suggesting that Africa's traditions are all the same. It helps us avoid reifying 'religion' as if it were a concrete institution, a thing apart from actual practice or separate from the living cultures in which they are embedded.

Meanwhile, Birgit Meyer (2020) argues that rather than abandon the term 'religion' in reference to what Africans believe, 'religion' should be processed and understood in relational sense between Europe and Africa. "My point is not that what is referred to as religion in Africa should not be designated by that term and be replaced by indigenous alternatives" (Meyer 2020, p. 159). However, she contends that employing the term religion in reference to what Africans believe makes their belief open to the outside world. After all, it is not only African scholars that study African indigenous religion, despite the (mis)representation that might have taken place at the turn of the European colonialism.

Having accepted the politically-motivated phrased religion status, it was then pertinent for most of the early African scholars and Christian converts schooled in European methodologies and philosophies to laboriously take up the challenge to provide persuasive arguments for the existence of religion in Africa, an attempt that has again become the ground for contention for secularism in the continent. The pervasiveness of religion in Africa also became an issue that had to be addressed later. As Landau (2015) recounts in the case of Southern Africa, which is true in most other parts, through the instrumentalities of law and education, a systematic religious emasculation called Western civilisation was established in order to disestablish African religiosity. In essence, rather than the law and Western education process the mind of the Africans to be secular, they apparently ended up making them more religious. The missionary educational system was not designed to free their minds or make them disenchanted with religion nor properly equip them to gain profitable jobs as the colonialists. It was meant to replace ATR with Christianity, and accept colonialism as civilisation rather than exploitation. By these instruments amongst others, the (Southern) Africans were groomed to appreciate the values of Christianity that

kept them focused on heaven, praying with firmly closed eyes, whilst losing their land and secular spaces to the colonialists. The colonial Christianity and by extension the Pentecostal brand that would later result and now proliferating furiously across the continent, are pitched in the contents of "The Slave Bible," contented only to allow a predetermined access to selected verses that glorified master-slave relations rather than freedom of the will (Igboin 2019b).

According to John Mbiti (1969, p. 2), for instance, "Because traditional religion permeated all the departments of life, there is no formal distinction, between the sacred and the secular, the religious and the non-religious, and between the spiritual and material areas of life. Wherever the African is, there is his religion." The idea of notoriety and incurability of religion as expressed by both Mbiti and Idowu (though they inherited the descriptions from a rather controversial origin) has hardly been refuted empirically, if anything, the presence, acceptance and expansion of Christianity and Islam amongst other missionary religions have concretely expressed it. Africans being religious in the contemporary sense encapsulates the reality of the presence and embrace of other religions. However, that is not the whole argument as it will be shown shortly.

Ukah (2016, p. 530) corroborates Mbiti's assertion when he avers, "in a traditional African society there were no atheists and religion suffused the social body—-and thus there was and still is a positive bias towards religion, and a visible relationship between humans and the transcendental world." Even though many African countries politically claim to be secular in post-colonial era, the reality on the ground tilts towards religion regulating much of the everyday experience of the people. The heavy reliance on religion is not just that the people are incurably religious (Idowu 1973) even in post-colonial era as it was in precolonial experience (Mbiti 1969), it can be argued that political and socio-economic reality of post-colonial policies have driven religious consciousness deeper into the mind of majority of Africans. Ukah (2016, p. 531) further notes that "several decades of military dictatorships in Nigeria, during which politics and civil society were constrained, literally forced many into religious activism, the church providing the only safe space for civil-society events and resistance." The imposition of structural adjustment programme by the World Bank and International Monetary Fund and its adverse consequences on the African masses gave vent to religious consolation and flourishing, particularly the Pentecostal brand, which tapped into the religious cosmology of the people. African Pentecostal Christianity does not only rely on the reality of the indigenous African cosmologies to thrive, but also the failure of post-colonial government to improve on the welfare of the citizenry. Thus, the preaching of prosperity gospel, which gives hope, whether psychological or otherwise, became a logical flow from the excruciating economic meltdown.

Curiously, Nelson Shang (2016, p. 166) argues that religion in Africa has been consigned to the dustbin because of the economic development in the continent: "it is undeniable that this influence of religion on public life has declined and that modern civilization, governed by economic values has produced a mind closed to the transcendent values." Shang interprets his assertion to be the effect of Western secularisation, which he thinks has shirked religious penetration into public and private spheres. He builds his submission on the Western premise that the effects of secularisation were irreversible, and wrongly contextualises it within the African setting. Shang's argument centres on the level of morality in public sphere, because according to him, the moral quotients in post-colonial Africa have drastically reduced compared with the highly religious moral life in pre-colonial Africa. He seems to maintain that there was a direct relationship between high morality and religiosity in pre-colonial Africa. The loss of public morality, he posits, is due to the infiltration of colonialism and missionary religions. Whilst it is indisputable that colonialism and missionary religions affected African reality in fundamental ways, it is contestable whether religiosity has reduced due to Western religions and economic programmes for Africa. The point is that religion has continued to determine how Western influences are interpreted and applied in many African societies. The reason is that most Africans have appropriated the values of these missionary religions into their lives, and where they fail to

resonate with their personal or group goal, they resort to the indigenous religious values and means to achieve their purpose. It is therefore not correct to argue that religion has declined in Africa.

However, according to Platvoet and van Rinsum (2003), Okot p'Bitek was the first African scholar of religion to vehemently and rationally question the empirical and logical basis of the incurability of African religiosity in the sixties and seventies. p'Bitek called the assertion a myth, because it was not engaged dispassionately before it permeated much of the academic study of religion in Africa and became a reference for the absence of semblance of secularism. As Platvoet and van Rinsum (2003, p. 1) argued, the myth of religious incurability "is a masterful counter-invention against the numerous European 'inventions of Africa', from classical times till now." They unfurl that had the African liberal Christian theologians known the negative implications of it, they would have carried out better to avoid it with caution. They contend that the struggle for independence and conversion to Christianity and the status it conferred could have misled some of them to believe that they were doing justice to Africa by influencing the African religious incurability discourse.

Platvoet and van Rinsum further challenged the religious incurability thesis, arguing that the promoters of the discourse had hidden agenda. Similar to every other human society, it will be strange to maintain that all Africans were religious in pre-colonial era. Although no empirical evidence is produced to the contrary, Platvoet and van Rinsum added that there was also no empirical evidence to substantiate the claim of everyone being religion. Their point tends to align with the highly religious society of ancient Israel, but the psalmist (Psalm 14:1 2015) would still recognise that there existed a 'fool' who thought that there was no God. That would later constitute the premise of St. Anselm's ontological argument for the defence of God's existence. In the same vein, it can be conjectured or speculated that there could be some Africans who could have been religiously indifferent or rejected the idea altogether.

Despite the overarching permeability of religion in social and moral life of pre-colonial Africa, "'there are people who reject religion . . . in . . . African cities', and also 'the loosening of religious ties is evident in Africa'" (Parrinder 1969, p. 231). Although Platvoet and van Rinsum would quote Parrinder with approval, the question of empirical substantiation of this claim is still contended (Olabimtan 2003). Whilst Gez et al. (2022) also make the point that Mbiti's position could erase the possibility of religious nones in pre-colonial Africa, they blame the Europeans for imposing their religious sentiments on Africa. According to them, "the idea of religion in Africa as an essential identity element and an unquestioned fact of life frames absence of religion as an oddity, and may help to explain not only the low rates of identification with the category of nones, but also the deep resentment that many Africans feel towards it" (Gez et al. 2022, p. 56). Had the European colonialists and missionaries have positive disposition towards African indigenous religion, they argue, Africans would have been able to publicly claim their religious status even today. The non-recognition of their religion inevitably made them stick to theirs as a form of resistance. However, they conclude that it would be difficult to confidently maintain that there were no Africans who did not subscribe to any form of religion no matter the low density of their numbers.

According to Platvoet and van Rinsum (2003), even Mbiti himself was not deeply satisfied with African indigenous religion as his critiques of it have copiously exemplified. Mbiti (1969, pp. 15–23) did not see a future in it even though he claimed to see its past, suggesting that the African indigenous eschaton is short-sighted and limiting. He also did not agree much with the ethical-spiritual relationship between God and man, just as he expressed reservation on the religion's anthropocentric goal. Mbiti would thus concede that the African indigenous religion was at best an equivalent of the Old Testament eagerly awaiting a New Testament messianism, fulfilled in the advent of Christianity. Olabimtan (2003) has raised objections to Platvoet and van Rinsum's claim, and insists that Africans were/are religiously incurable, in support of Mbiti. The attempt by Platvoet and van

Rinsum to forcefully insert secularism into African religious universe, Olabimtan argues, is not only self-serving, but also resonates with what Adogame (2022, p. 20) refers to as "some Western (European) captains of academic industries who often claim to know Africa better." Even though Platvoet and van Rinsum (2008) would reply to Olabimtan's (2003) objection to their claim, their objection focused more on methodological defence of their position than the substance of religious incurability. The assumption is that once the methodological framework is sound, the conclusion should seamlessly follow. According to Nye (2019), such conclusion can be engaged more rigorously because it assumes that methodology and theory are in themselves neutral. Nye argues that methodology and theory do not evolve in a vacuum; they have context and goal. As experience has shown in the case of Africa, they are schemes of power relations between Western and African scholars on African history, religion and philosophy (Igboin 2022b).

Upon deeper reflection, one can argue that essentialising African indigenous religion by Protestant liberal theologians in their pan-Africanist agenda also raises further questions in addition to those earlier raised by early Western anthropologists about the existence of 'religion' in Africa. If African indigenous religion was not regarded and accepted as theologically capable of ultimately saving its adherents as Christianity, it shows that Africa might not have had 'religion' in the real (Western) sense of it. After all, why have a religion that cannot save you! The flip side of the argument would be that if as early Western anthropologists claimed that they did not find 'religion' in Africa, was there secularism in Africa prior to the advent of missions and colonialism? Put differently, why ask for secularism in Africa if there was no religion *qua* religion in the mode of Christianity, which is represented as the standard gauge of religion? Before we respond to these questions, it will be pertinent to cede with Nongbri, and from the trajectories of 'religion' as follows:

> . . . that religion does indeed have a history: it is not a native category of ancient cultures. The idea of religion as a sphere of life separate from politics, economics, and science is a recent development in European history, one that has been projected outward in space and backwards in time with the result that religion appears now to be a natural and necessary part of our world. (Nongbri 2015, p. 6)

Since religion has a history, it was created and conceptualised at a point in time. In addition, as it has been made clear, it became a veneer of power relations between the colonisers and colonised in Africa. From the foregoing, it can be argued that Africa has always had religion—the consciousness, belief and worship of God. However, it needs to be stated that permeability of religion did not rule out the possibility of religio-secularity. The denial of the existence of religion by the Europeans and their subsequent admission of its existence were politically motivated, which have severe implications on both Africans and Europeans, especially in the search for secularism in Africa. The next section will examine the presence or otherwise of the phenomenon of secularism.

## 3. Scramble for 'Secularism' in Pre-Colonial Africa

For some Europeans, secularism is not considered as an accident of history but as "an achievement of civilization" (Engelke 2015, p. 5). What does this imply? "Therefore, secularism as a political project only makes sense in relation to *a certain kind of society*— those with 'world religions,' not just witchdoctors or spirit mediums" (Engelke 2015, p. 5, emphasis added). This definition is not problematic to comprehend because it clearly inflects that secularism can be conceived in terms of European exceptionalism, one that has the Christian projection as its background. Attempting to sniff for secularism in pre-colonial Africa that had no religion, let alone world religion, in the estimation of the West, ironically constitutes a momentous waste of time. Another takeaway is that secularism is not a universal concept, but however, through an evolutionary process, a particular society can achieve it as a fallout of civilisation. It is also a scheme of power relations between the coloniser and the colonised, at least, in the African context.

Before we pursue that argument further, it is helpful to nuance the valences associated with the concept which are indisputably Western in history and epistemology. From the

sociological point of view, secularisation is concerned about how religion empties itself or is divested of its grip on social life and carves for itself an independent ontology and epistemology. In addition, it means the decline in public religious participation, religious membership, and increased privatisation of religion (Casanova 1994). Although there is no uniform secularisation thesis, the three major characteristics that run through the thread are the ones highlighted here: decline, privatisation, individualisation of religion and institutional differentiation of modern society (Onishi 2018, p. 3). Institutional differentiation means that institutions of modern society constitute separate social, religious, economic domains. These domains were not completely independent of religious order in European pre-modern period. Despite the merits of secularisation thesis, many of its proponents have abandoned it as unrealistic in post-modern European society. Religion, as they have come to realise, is "furiously" back in the public domain as it had always been (Berger 1999; Cox 1995). It is therefore not surprising that Jeffrey Hadden argued that "secularization theory had not been subjected to systematic scrutiny because it is a doctrine more than it is a theory" (Onishi 2018, p. 4). As a doctrine, it was preached without corresponding empirical data to substantiate it as a theory. As it has turned out to be true, the characteristics of secularisation 'doctrine' as outlined above are not in most of African countries.

On the other hand, "Secularism refers more broadly to a whole range of *modern secular worldviews* and ideologies that may be consciously held and explicitly elaborated into philosophies of history and normative-ideological state projects, into projects of modernity and cultural programs. Or alternatively, it may be viewed as an epistemic knowledge regime" (Casanova 2009, p. 105 emphasis added). The stress on modern secular worldviews is immediately suggestive of its antonymy: pre-modern secular worldviews. It can be assumed that there is a difference between the two conceptions of secularism historically even in Europe. In any case, secularism as Asad (2003) will want to know, is a modern ideological project geared towards freeing the modern society from extraneous or religious impulses. This idea of emancipation from irrationality and faith undergirds secularism's project. Reason or rationality is the masterstroke of secularism. Any human domain that reason cannot penetrate is deemed irrational, and religion, according to this argument, is a quintessence of irrationality. Even though secularism projects a non-religious vision, it is not free from non-theistic assumptions such as reverence, ethics and so forth. However, whether this project is realistic or not is also open to critical interrogation, because according to Margaret Canovan, secularism is a "project to be realized" and built on "a foundational myth" (Onishi 2018, p. 8). The main point of contest for us in this paper; however, is that secularism tends to subvert religion and its claim to omnipotence, but favours calculative thinking. What is subverted in religion is ascribed to the human self. This humanist atheism projected in secularism is objected to in Africa.

In addition, Burchardt et al. (2015, p. 10) argue that secularism is an "ideological-philosophical program of separation, and—by contrast—use the concept of secularity as an analytical term for the culturally, symbolically, and institutionally anchored forms of distinction between religious and non-religious spheres and material spaces." This "ideological-philosophical program of separation" did not take place in a vacuum or seamlessly in the history of its evolution.

The Wars of Religion fought between Catholicism and Protestant Christianity were largely based on the interpretation of God. The conflicts were result of the Protestant Reformation that hinged on how best to attain salvation. The irreconcilable positions maintained by the Reformation teams against the Roman Catholic caused the 'secession' of the former. Acquisition of political power that resulted from the Reformation by the different groups led to the conflict that would last for decades and disrupt social and economic life. In order to creatively bring about a permanent truce, it was incumbent on thinkers to tease out how best to make both sides understand the devastating effects of the wars on social life and economy. As John Locke would suggest, it was needless to concentrate on argument about which strand of Christianity was true, that argument could be confined to private sphere, but the focus should be loyalty to the legal codes that

would ensure the safety and development of nation-states that belonged to all irrespective of sectarian persuasions. Locke's ideological-philosophical programme of separation of religion and politics would later be intellectually amplified and projected to keep the institutions of religion and politics apart (Nongbri 2015). However, it was these sectarian or provincial debate and strife that were later extended to the world through colonial contacts. This intra-Christian sectarian division, according to Akinwumi (2008), reflected in the scramble for Africa, as different European nations 're-worlded' and redrew the religious cartography of Africa in accordance with their sectarian traditions and different nuances of secularism. Within the divided Christianity also lied different notions of secularism.

Following the arguments of Burchardt et al. (2015) and Kleine and Wohlrab-Sahr (2020) in their groundbreaking projects on multiple secularities, we find a theoretical anchorage to assume that secularism is neither fixated in the West in all its projections nor excluded non-Western societies from capable of inventing it. Secularism is thus a process rather than a finished product. In order to circumvent the fixated meaning of secularity as connoting "impermeable separation" for example, Kleine and Wohlrab-Sahr (2020, p. 14) aver:

> we now understand 'secularities' as interrelated epistemic and social structures, in which the religious and the nonreligious are socially differentiated (institutionally, legally, organisationally, spatially, habitually, lifeworldly, etc.) and are conceptually distinguished (taxonomically, semantically, discursively, symbolically, etc.) by relevant actors in a binary schema, whereby the corresponding demarcations can be variable, negotiable, controversial, and blurred. We are therefore combining a formal analytical concept with an interest in structures of conceptual distinction and social differentiation that can be identified empirically (emphasis in original).

The willingness and openness to explore the possibilities of the presence of secularism in non-Western societies is salutary because it has the potential to argue for its universality as well as dispossess it of the civilising proclivities that have made the concept suspect and exclusive. Even though Rijk Van Dijk (2015) argues that studies on secularisation is at its early stage in most of Africa, he also notes that its Western "essentialized differentiation" has not helped in co-production and 'co-constitutiveness' in other climes largely because it is presented to them as a finished product. This is hinged on the now controversial assumption on the absolute religiosity of Africa, a description we have argued earlier was a politically motivated strategy to achieve a predetermined purpose. However, as Landau (2015) puts it: "Africans' politics were marked as religious in contradistinction to the overrule of Africans by whites—which proliferated a field of secularity, one of bureaucratic memory and 'expertise.'"

Leo Igwe (2017) argues that the 'foreignisation' of secularism and opposition to it in Africa are both foreign both theoretically and empirically. Igwe notes that secularism cannot be an exclusive concept to the West, although the neologism and degree of attention paid to it may be conceded to the West. According to him, what is critically important is to dispassionately engage in ethnographic study of communities in Africa beyond the remit and reach of armchair anthropologists who created their own image and version of Africa for self-serving purposes, much of which has been vigorously challenged. The vestiges created by most of those early scholars on Africa and her reality were geared towards political end: the justification of colonialism back home in the West rather than accepting the truth as it was on the ground. Thus, Igwe argues that it was the West who propounded the absence of secularism in Africa just as they, through local compradors, now agitate for secularism in Africa.

In addition, Igwe makes that point that even though the West flaunts secularism as a civilising agenda, they certainly did not pragmatically display it. For instance, the activities of the Christian missionaries and colonialists were hardly distinguishable from each other. The collaboration between missionaries and colonialists, the support of the latter for the former in achieving their goals, and sometimes restraint against the former in support of Islamic cause, all contributed to the position that the colonialists did not practically

separate religion from politics. If anything, the colonialists used religion as a veritable tool to achieve their political goals. In Igwe's own words, "It is important to note that despite the proclaimed secular nature of the colonial state, there was a mix between colonial politics and colonial religion . . . [such that] [Africans] did not know who came to missionize and who came to politicize" (Igwe 2017, p. 26). The blurring of the operations of the mission and colonialism could have possibly been responsible for not noticing a functional form of secularity at place. As Igwe further notes,

> Prior to the colonization of Africa, various empires, kingdoms and chiefdoms existed in the region and were administered based on models of indigenous governance. Different relationships existed between the shrine and the state, the religious order and the governmental order, the sacred and the secular. There were informal secular approaches to governance within these kingdoms and chiefdoms despite a lack of codification (Igwe 2017, p. 27).

However, Igwe did not define what he means by secularism, and we argue that he does not mean secularism but secularity as his ethnographic accounts—on Igbo and Dagomba, which will be analysed in the next section—have shown. Igwe's intention is to demonstrate that in pre-colonial Africa, there were elements of separation of institutions, but he does not deny the existence of God or absolute reliance on reason in the administration of the societies. However, what is unarguable is that colonialism disrupted African political system just as Christianity's and Islam's agonistic relationship to African indigenous religion suffered some setback. Colonialism and missions whether Islam or Christianity might not have introduced secularism into Africa; what can be observed is that these forces compelled majority of Africans to change their loyalty and reverence from their indigenous deities to Western deities. In addition, where resistance to these incursions was put up, the colonialists, without condemnation from the missionaries, militarily invaded the African kingdoms as in the case of the Benin Kingdom in February 1897 (Osadolor 2011).

From the foregoing, we can make the following deductions: secularism, secularisation, and secularity are Western concepts developed in reaction to the prevailing context in the West. We then argue that these concepts were not in pre-colonial Africa as they are now understood. What was prevalent in pre-colonial Africa is religio-secularity. Religio-secularity is the concept that states that social institutions might be differentiated, but they were not entirely independent of religion. In addition, religion provided the formidable basis for the understanding and interpretation of social institutions. However, it must also be noted that religion itself could not survive and maintain its powers without the secular institutions such as agriculture, economics, and so on. As the Igbo societies have shown above, and Wariboko's (2022a) study of the Kalabari people of the Niger Delta of Nigeria also corroborates, any spirit whose demands became burdensome on the people could be denied worship and gradually made to fizzle away. This social or secular construct of religion was not however interpreted to denigrate the overarching presence of religion or to promote other institutions above it. Both religious and secular institutions did not deny the existence of God as in secularism; they did not privatise or individualise religion as in secularisation; and they were not completely separated from influencing each other as depicted in modern secularity. Whilst there was room for individual expression, fulfilment or freedom, the community played a prominent role in individual life and human flourishing (Ogude 2019; Wariboko 2022b).

We agree with Burchardt et al. (2015, p. 8) that "while the initial impulse in some world regions to engage with secularity may have happened through colonial or imperial encounters, more important was the question of what kinds of ideas and practices were eventually registered within the vernacular categories that were created." In exemplifying the presence or absence of secularism in pre-missionary and pre-colonial Africa, it is pertinent to state that different ethnic nationalities had different political ideologies and administrative systems. Whilst some were stateless or acephalous, others had developed a monarchical system. These various political systems affected in major ways the conception of religiosity and secularity. Avalos (2018, 2020) provides us a veritable lead

here. She argues that ethnic studies should be used as methodological and pedagogical approaches to decolonise the production of knowledge, because ethnic narratives provide better and nuanced understanding of a people. Ethnic studies also become critical because it challenges the colonial omnibus and hegemonic accounts and narratives about Africa. The representative examples below follow this Avalos' view. The authors used ethnographic methods to generate their data, which we analyse to justify the presence of religio-secularity claim in pre-colonial Africa. In selecting these examples, I take cognisance of Benin Kingdom being a highly organised political system long before the advent of missions and colonialism. Igbo, Dagomba and Jola were mini-states, less central than the Benin Kingdom. The Dagomba and Jola had encounters with Islamic traders and jihadists, whilst Benin and Igbo had encounters with Christian missionaries before colonialism. The Jola were colonised by the French, whilst the Benin, Igbo and Dagomba were colonised by the British. Even though the British and French colonial administrative styles were different, we observe a similar reaction to colonialism and missions, and the attempt to replace the public role of indigenous religion in post-colonial era by Christianity and Islam. This attempt has been a major cause of inter-religious violence, especially in Nigeria.

## 4. The Pre-Colonial Igbo and Dagomba Societies

The Igbo ethnic nationality[2] in Southeast Nigeria had a mini-state, a decentralised political structure. Every family had its leader—the father—and a group of families formed a community. There was no kingship system as everyone was loosely regarded as a king in his family. The Igbo did not hold allegiance to a centralised authority, but every household established its own authority. Every autonomous community formed a social and political system based on gerontocratic succession (Onyeozili and Ebbe 2012; Onumonu 2016). However, in some parts of Igbo where there was kingship arrangement, which, of course, was largely attributed to the forces of migration, but still in pre-Christian and pre-colonial era, Igwe (2017) unfurls that the Eze (king) and Dibia (priest) had almost neatly differentiated and interdependent roles. Whilst the kings were charged with the daily administration of the kingdom, the priests poured libation and offered sacrifices to the deities. Whenever there was need for consultation with the deities for personal or community purpose, it was the responsibility of the priests to do so. However, the priests operated under the kings.

Igwe (2017) also reports that amongst the Dagomba of northern Ghana before the advent of Islam and British colonialism, the chiefs administered the community, whilst the priests were charged with spiritual responsibilities. These differentiated roles were neatly ingrained in their monikers: the chiefs were called and regarded as the owners of the land and the priests were addressed as the owners of the gods. However, there could be rare occasions when a chief could function as a priest; this provided example of mix of both political and religious authorities, and it was mediated by the community in order to forestall autocratic abuse of powers. As Igwe observed, the introduction of Islam to Dagomba did not significantly alter their perception of the roles of chiefs and priests; they now regard the Imams and mallams as part of the owners of the gods, thus, expanding the scope of religious order. This is despite the fact that Islam does not neatly differentiate between secular and religious order. Igwe thus compared the Dagomba with the Igbo and concludes that both societies recognised the separate but synergistic roles of kingship and priesthood prior to the advent missionary religions and colonialism.

In the autonomous communities, it would seem that the priest had incredible public prominence as he mediated between the community and the deity, but his position did not confer any absolute authority on him over the community. His continued retention of the office and relevance did not depend on his own will, but absolutely on the judgement of the community over which he presided religiously. Although the community had great reverence for their deity, on whom they depended on for bumper harvest in an agrarian setting, they also believed success in life was not conferred on anyone who would not deploy his skill and strength. "The pre-colonial Igbo life was highly materialistic society.

However, at the same time a strong spiritual dimension controlled its materialistic aspects. In addition, the culture was successful because there was a complementary balance between the two, the materialistic dimension of Igbo life being related to its masculine principle and its spiritual dimension being related to its feminine principle" (Onyibor 2016, p. 112).

## 5. The Benin Kingdom

It is important to point out here that the concept of state is mostly used in its Western analytical categories in reference to African pre-missionary and pre-colonial political system. In fact, many of the modern concepts or theories used to analyse African reality are products of the Enlightenment (Abraham [1962] 2019). Analysing the Benin Kingdom c.900 to 1897, Dmitri Bondarenko (2006, p. 31), using Weberian concept of state, has this to say:

> The administrative system of the Kingdom formed in its most important features during the 13th–mid-15th centuries and remained basically the same till the end of the country's independence [in 1960] (introduction of the title of Queen Mother— Iyoba in the 16th century may be recognized as the only important innovation in this sphere of the subsequent period). From the mid-15th century on, mostly a redistribution of functions and amount of power between the supreme ruler and titled chiefs, on the one hand, and among different categories of the chiefs, on the other hand, was taking place.

In Benin Kingdom, the Oba (king) of Benin is the political and spiritual head of the kingdom. However, this does not immediately mean that he performed all the ecclesiastical and prophetic functions or duties of the priests. Osadolor's (2001) thesis on the Benin Kingdom further challenges the assumption that there was no separation of authorities in Africa. Osadolor makes the point that in pre-colonial Benin Kingdom, there was separation between civil and military authorities on the one hand, and military and political authorities on the other hand. The structure of the kingdom was such that all chiefs did not have the same portfolios. In addition, the controversies raging over the origin of the kingdom also add another impetus to the relationship between priests and monarchs where the listing of the former differs from the latter with regard to the reign of individual monarchs. "The kinglist published by Jacob Egharevba, and the one collected from Isekhurhe, the priest of the royal ancestors do not agree with the lists published by European writers" (Osadolor 2001, p. 13). Egharevba's (1968) list has been alleged to have been influenced with European methodology.

Although the Oba of Benin had full executive powers, "the priest of Okhuahie was responsible for the state army. The *Ewaise*, a guild of 'traditional doctors' who controlled the shrine of *Osun-okuo* (war medicine) also played a prominent role" (Osadolor 2011, p. 201). Even though the Oba of Benin could make personal offerings to Osanobua (the Supreme Being), the priest received and delivered divine messages to the king (Welton 1969, pp. 101–2). Whilst the king may have general knowledge about spiritual things, the priests "have special knowledge about Osanobua" (Welton 1969, p. 102). However, in Aruosa Temple, it is the king that appoints the priest who officiates in public worship and carries out religious duties. In most circumstances, the priests and high rank chief would not be expected to be "kneeling as befits the dignity of their office and may sit even before the Oba" (Ojo and Ekhator 2020, p. 164).

## 6. The Pre-Colonial Jola Society of Senegal

In his ethnographic study carried out amongst the Jola of Casamance, Senegal, Olga Linares (1992) observed a proliferation of spirit-shrines with polyvalent functions. Linares argues that such belief and practice were built on cultural constructs of the people. Whilst some shrines existed to cause rain to fall, protect the crops, punish moral offenders, when the indigenous courts of law were still operational before the advent of Islam and colonialism, "a special spirit was in charge of punishing those who committed the heinous crime of murder" (Linares 1992, p. 25). Although she posits that

In societies where bureaucracies are missing and there are no standing armies, as among the relatively self-sufficient rural communities of Africa, religious beliefs and ritual practices often reinforce many aspects of political economy. Cultural ideologies and symbol systems usefully provide a legitimating idiom for the values and aspirations surrounding the economics of role behavior (Linares 1992, p. 15).

Even though Jola society can be described as politically acephalous, the gerontocratic system ensured some political order. The elders had incredible access to religious resources and exercised religious authority in the maintenance of their society. In Jola political system, anybody could become an elder and also exercise the authority conferred on elders. Although Linares, such as many other European scholars, expends long space explaining how religious order determined much of the social and moral order amongst the Jola, her own evidence suggests that that was not absolute or unquestioned. In other words, the pervading influence of the spirits in the Jola society and the symbolic residence of those spirits in one person did not translate to or result in the person's overarching exercise of power over moral, political and communal order. His authorities were wedged, limited, checked and balanced, not only by the religious order from which it was widely believed he derived his authority from, but also by the political order in the society he presided over. Schumacher's (1975) and Baum's (1987) earlier ethnographic analyses of the Jola show that there were limits and in fact, a sort of separation of powers, which was blurred by other earlier Europeans who undertook some study of the Jola. Linares (1992, p. 41) discovered that the Jola had a balanced political system that ensured that too many powers were not concentrated on either the priest-king or elders of the society. This check and balance system regulated the functions of both leaders for the social development of the Jola society.

Linares argued that the religious and political order of the Jola took a different turn when Muslims, and later French colonialists invaded their society in the 19th century. Jola people being warriors, resisted the incursions of Muslims who wanted to convert them into a theocracy and colonialists who wanted to politically and economically exploit them. The jihadist approach of the Muslims was a major setback for accepting Islam because the Jola were averse to religiously centralised authoritarian structure against their decentralised and dynamic system. For hundreds of years culminating in the 19th century, the wars between the pre-colonial Senegalese and Muslims shaped their relationships not only on cultural, ideological, political and social order, but also religious order (Glinga 1988). Moreover, "their armed conflicts with the French increased their suspicion of centralized government, postponed their acceptance of new economic endeavors and delayed their incorporation into the nation-state" (Linares 1992, p. 93). Although they would lose to both Islamic and colonial forces in the long run (Foley 2009), the French colonial authorities backed the Muslim invaders to rout the Jola, which would affect their cultural development.

The four cases examined above demonstrated that despite the claim of African religious incurability, the societies were organised politically to check the excesses of both religious and political institutions. The Igbo and Benin in Nigeria were defeated by the British colonial authorities, but the indigenous religious and political systems had been exposed to Christian missionaries. For the Benin kingdom, Christian missionaries had in the 15th century unsuccessfully persuaded the Oba (King) of Benin to be converted to Christianity. Even though the missionary efforts died then, Benin and Portugal had exchange of ambassadors. It was the British that would eventually conquer the kingdom in the 19th century, whilst missionaries succeeded in planting Christianity. Even today, the Benin and Igbo still practise their religio-secular political system.

In the case of Dagomba and Jola, the Muslims invaded them and attempted to obstruct their religio-secular structure. Whilst the Dagomba seemed to have absolved the Islamic deity as part of the owners of the gods, the British would further strengthen the Islamic cause through an indirect rule. The Jola's case was a bit different because they protested for over four centuries in order to maintain their religio-secularity. However, the French

supported the Muslims to defeat them, and the policy of assimilation adversely impacted on their religio-political system.

Today, although Senegal constitutionally claims to be a secular state, it is glaring that religion still permeates and plays significant role in both public and private spheres. Its enviable global position on secularity and tolerance radar is worthy of emulation, but also requires a deep historical and nuanced analysis, without romanticising the past. Islam has penetrated the Senegalese country so deeply that erasing its relics from public sphere is difficult, if not impossible. The point is that regardless of the avouched status of Senegalese secularity, Islam has entrenched itself most vigorously in state institutions, politics and policies. The distinctiveness of secularity in Senegal is that there exists a kind of "social compact" which makes for tolerance despite the fact that Islamic values have penetrative impact, respected and largely define Senegalese national identity. The uniqueness of Senegalese secularity also reflects in the fact that although the country adopted the French style of secularism which entails separation of religion from government, the reality shows that there is a mix between religion and religion. Senegalese secularism is avidly concerned about freedom to practise one's religion, recognition of traditional values and pluralism. In other words, government involvement in religious affairs should not be to the detriment of other religious groups or in favour of a particular religion. This variant of Senegalese secularism raises critical questions amongst the elite who are committed to secularity and its values, but also have to contend with the Islamic values in the public. The knack to manage this complex and competitive nature of secularism has largely given credit to Senegalese social compactness (Herzog and Mui 2016).

In any case, kinship system continues to accentuate the level of tolerance in Senegal (Stepan 2012, 2013; Smith 2013); a kind of tolerance that they hardly traced or ascribed to the nature of the pre-colonial communal system that pervaded their society. Even though Stepan (2013) recognises the fact that French type of secularism is totally hostile to the religious nature of the Senegalese indigenous society, no attention is paid to it. In his words, "the clerically hostile form of French laicite does not actually apply to state-religion relations in Senegal" regardless of the similarities in both countries' constitutions (Stepan 2013, p. 216). The unanswered question has been that the uniqueness of Senegalese secularity that respects multi-religiosity of the Senegalese society must be traced beyond French secularity adopted by Senegal and Islamic form of values, but to the contestation and absorption of Senegalese pre-colonial religious diversity which is the fulcrum of modern tolerance.

These examples clearly show that the idea of secularism is far-fetched. Chidester (2012) argues that what the European called "wild" religious practices of the pre-colonial Africa constantly re-emerge and calcify the premise of religio-secularity. Jacob Olupona's (2011) and Adogame's (2022) area and ethnic studies have further solidified the argument towards religio-secularity rather than secularism. As Adogame (2022, pp. 7–8) found amongst the Oza people of Edo state, "The perception of religion as a phenomenon separate from culture is not a suitable reflection of the embedded nature of 'religion' in African cultures."

## 7. Concluding Remarks: Looking Backward and Marching Forward

It is clear from the foregoing that the ideological tapestry of Africa during colonialism and post-colonialism was influenced by three critical planks: (a) African religious and cultural heritage and values, stubborn yet supple in the face of (b) Islamic incursion that added to the African religious universe already dominated by powers, and then Christianity that added educational impetus to its project of conversion; and (c) colonialism with its imperialistic tendencies poised to redraw both physical and religious map of Africa, causing tension between and amongst religious traditions and diversities within them. At the centre of those tensions were Africans on the one hand, and the struggle to build a secular state from the rubric of colonialism, on the other. However, if the post-colonial Africa will succeed in building a virile secular state, two basic questions need to be addressed.

One, was there religion in pre-missionary and pre-colonial Africa? Two, was there secularism in pre-missionary and pre-colonial Africa? The first question has largely been

answered by many Western scholars in the colonial period, namely: upon deeper look, the conflated beliefs, traditions, animisms and so forth eventually translated into a kind of phrased religion, peculiarly African, and unsuitable to be categorised as world religion in their own estimation. Although "African Traditional Religion" or courses in it can be studied in some Western universities; African Studies Associations regularly hold conferences to discuss and interrogate African indigenous/religious thoughts, African Futures, traditions, beliefs and contemporary realities in Africa, and grants are being awarded to undertake 'sky-breaking' research projects on Africa and her realities, ATR is yet to be formally recognised as a world religion. It is still being treated as an object rather than a subject by some Western scholars. This makes us to tweak the question if ATR is a religion without calling it a religion in the Western sense. Although only a few still premise their reference to ATR in the cloak of the armchair anthropologists at the turn of colonialism, the need to recalibrate the study of ATR as a popular and academic course as other world religions urgently beckons on (African) scholars.

However, since in the course of time ATR was 'discovered' or 'invented' to be a religion, its permeability in all forms of life might have made it difficult to recognise characteristics of secularism. As it has been argued earlier, there was no empirical evidence to prove the religious incurability thesis whose origin is not only controversial, but also strategically created to serve some political end (Platvoet and van Rinsum 2003). Neither has there also been salutary empirical evidence to prove otherwise (Olabimtan 2003; Wijsen 2017). This clearly makes it difficult to talk about secularism in its Western sense in pre-colonial Africa. However, it would be helpful to re-image and re-imagine the pre-colonial African era through the lens of religio-secularity. In this way, we are not using the contemporary knowledge of secularism to adjudge the pre-colonial era.

The recognition of the vagaries and polarities of life is one critical arena to asseverate a religio-secularity understanding of social communal existence in pre-colonial period in response to the second question. For instance, the belief in the presence and efficacy of ancestorship is hinged on the experience of an ancestor whilst alive will be useful and applied now if and only if it is relevant and positive. In fact, that is why becoming an ancestor depends on the kind of life someone led rather than the amount of sacrifices he offered. In other words, even though ancestors are believed to be the custodians of morality, they must have lived a morally worthy life on earth to act as moral guardians as ancestors. That is why it is not all dead people that become ancestors eventually (Igboin 2022a; Louw 2019; Clark 2012). Here, a decision is made, and personal responsibility for one's action is established. Almost all Africans were faced with this fact of life. In addition, pre-colonial Africans lived with the messiness of the finitude of life: salvation was not a journey to exit this world, it was to live productively in it, and return to it even at death. This present world, and not heaven or hell, is the drama-scene: how to improve on it pre-occupied much of their thoughts without disregard to the supernatural, hence laws, taboos, regulations and their enforcement. This is captured by Nimi Wariboko (2022a, p. 65) thus: "*Bibibari* ['recanting' amongst the Kalabari, Niger Delta region of Nigeria] is a reorientation of a particular life toward the future that is not certain and is still bounded by a retained past that has value, and a distended present is between them (past and future)." Here lies the religio-secularity vision of pre-colonial Africa.

From the political perspective, Kwame Nkrumah the first indigenous President of Ghana amongst other African nationalists was confronted with task of building a secular state. For Nkrumah, an African secularism should not be constructed on the metaphysical rationalism or divine revelation that Christianity and Islam claim, but rather on the pluralism inherent in African Traditional Religion as well as the presence of the missionary religions. Accordingly, he avers,

Insistence on the secular nature of the state is not to be interpreted as a political declaration of war on religion, for religion is also a social fact, and must be understood before it can be tackled. To declare a political war on religion is to treat it as an ideal

phenomenon, to suppose that it might be wished away, or at worst scared out of existence. (Nkrumah 1964, p. 79).

Nkrumah realised that Ghanaians, and by extension, Africans have strong attachment to their pre-colonial religious values and commitment to political leadership. Disrupting this synergy in a bid to pursue 'rugged' secularism, he believed, would spell doom for economic modernisation. He quite understood that economic philosophy does not just pursue rational arguments. As Wariboko (2022a, p. 178) clarifies it, "economic philosophy is neither economic nor rational; it is religious and ideological." In Africa, it must involve a synchrony of communalism and individualism. For Nkrumah therefore, there must be ingenious ways to construct a secularity that is at once development-oriented and African. Put differently, the project of non-atheistic secularism will be peculiarly African, which we term religio-secularity. Of course, such vision of secularity appears not to have practical resonance because, according to Ebenezer Addo (2020a, 2020b), Nkrumah did not seem to appropriate Christian and Islamic values into his notion of non-atheistic secularism. With the political influence the two missionary religions had garnered, religious tension was bound to occur. In addition, religious tension does occur aplenty, and it borders on whose God is superior: Judeo-Christian, Islamic or African! In sum, the search for secularism in pre-colonial Africa will still be on for some time in the future, but the religio-secularity is more appropriate to describe the African worldview before the advent of missions and colonialism. Furthermore, as the reality on the ground has shown in most post-colonial African countries today, religio-secularity is avidly in practice, despite its challenges.

**Funding:** This research received no external funding.

**Institutional Review Board Statement:** Not applicable.

**Informed Consent Statement:** Not applicable.

**Data Availability Statement:** Not applicable.

**Conflicts of Interest:** The author declares no conflict of interest.

## Notes

[1] His Arab name was Al-Hassan Ibn-Mahammed Al-Wezaz Al-Fasi.
[2] Ethnic nationalities were independent ethnic groups that were forcefully amalgamated by colonial authorities to form a nation. The Berlin Conference of 1884–1885 altered the cartography of ethnic nationalities in Africa by partitioning and distorting the organic ethnic system in the continent. In the case of Nigeria, the Igbo people largely have a common identity and values. But in 1914, the British colonisers amalgamated different ethnic groups that were earlier operating as independent peoples to form what is called Nigeria today. Over a century later, these ethnic groups—Igbo, Yoruba, Edo, Hausa, and so forth still believe that they want to go their separate ways, hence the agitations by the Indigenous People of Biafra (IPOB), the Yoruba Nation, and so forth (see Osaghae and Onwudime 2007; Usuanlele and Ibhawoh 2017).

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
