# Peer review of "The Scramble for Religion and Secularism in Pre-Colonial Africa"

_religions, doi:10.3390/rel13111096_

Round 1

Reviewer 1 Report

The authors ask a very valid and important question: How to explore and think about religion and secularism in pre-colonial Africa whilst considering the vexed colonial history of both categories? In the authors’ own words, the question reads as: ‘Why ask for a certificate of secularism in Africa if there was no religion qua religion in the mode of Christianity, which is represented as the standard gauge of religion?’ (p.7). Accordingly, the authors insert their text into the debate about ‘religion’ in Africa and argue that this debate should also keep an eye on ‘secularism’ in Africa. This point is well taken and makes an important contribution to the debate.

The authors unravel the history of the complex debate about religion in Africa and excavate the largely unacknowledged biases and normativity of the concept which allows them to highlight the problems that the analytic application of the Western conception of ‘religion’ to African religions entails – namely, the strange status of ‘ATR’ in Western thought. Whilst Africa was originally considered to have no religion by Western observers, this came under challenge in late colonial times. Different scholars proposed the thesis of Africans being incurably religious (Parrinder, Mbiti, Idowu; Olupona today) as they perceived religion to be pervasive of and omnipresent in the lives of people on the continent. As it seems, the authors side with this thesis as they also write that ‘Even though many African countries politically claim to be secular in post-colonial era, the reality on the ground tilts towards religion regulating much of the everyday experience of the people. The heavy reliance on religion is not just that the people are incurably religious (Idowu 1973) even in post-colonial era as it was in pre-colonial experience, it can be argued that political and socio-economic reality of post-colonial policies have driven religious consciousness deeper into the psyche of majority of Africans.’ (p.5). Therefrom, the authors ask what to make of secularism and how to trace it in pre-colonial Africa. Obviously, Africa being all religious, secularism and its place in African societies becomes a question.

Drawing on the work of Casanova and the Multiple Secularities Project in Leipzig, the authors define secularism as ‘a process [of differentiation] rather than a finished product’ (p.8). They then set out to study how such processes of differentiation (mainly between the state and religion) occurred or played out in pre-colonial West African societies. Glancing at differentiation processes in Igbo, Dagomba, Benin, and Jola societies, the authors highlight that such processes did indeed exist in pre-colonial West African societies. Therefore, these societies can be described and considered as having ‘secularism’ albeit in different forms and processes than in the West. In the Conclusion, the authors reflect on how looking back at African secularism might help African nation building projects in the present and future. To this end, the authors suggest the notion of a ‘religio-secularity vision’ and ‘the project of non-atheistic secularism [that] will be peculiarly African’ (p.15). According to the authors, ‘the religio-secularity seems more’ (p.15), and the essay breaks off in the middle of this sentence.

On the one hand, the essay develops a pertinent and promising question. On the other hand, its argument and narrative are not yet well-developed, and its points are rather asserted than substantiated (the word ‘fact’ is used repeatedly in the text, as I find myself wanting for substantial demonstrations – there is a lot of ‘telling that’ in the text but not enough ‘showing how’). Furthermore, the text reads more as a partly incoherent literature review and not yet as a compelling discussion of the authors’ main argument and points. There are a lot of ideas on the table, so that I found myself occasionally at loss in the text. Overall, I fully agree with the authors’ question and some of their critiques (namely and amongst others: the problems resulting from conceiving religion in the image of the West and then ascribing it to Africa, their objection to Africa being perceived as non-secular), but I find myself disagreeing with their conclusions and remain to be convinced of several central points (namely and amongst others: Africa being incurably religious, ethnic studies as peep into the past, religio-secularity vision as peculiar to Africa). The paper needs some major revisions.

My main suggestions for the revisions are:

-        Include a discussion of your methodology and operationalize your two key terms (religion, secularism). How can we study both in pre-colonial Africa? Some answers to the question are already found in the text, but it would benefit the reader if this was discussed and clearly spelled out in the Introduction.

-        How do you understand and use the categories religion and secularism in your text? Add working definitions early on. I found it very compelling to consider secularism as process, but I would have benefited from you stating clearly in your own words what this process is or looks like.

-        Your text oscillates between a discussion and revaluation of ATR and your search for secularism in pre-colonial Africa. I’d leave the ATR debate be and focus more on the secularism line of your text.

-        What is your point, goal, or argument? This should be stated clearly in the Introduction. I suspect that you wish to revisit secularism in pre-colonial Africa to provide present-day African states with ‘African’ tools or ideas to build their own secularism.

-        Is Africa indeed as incurably religious as the authors suggest? On what grounds can you make such a claim?

-        Why pre-colonial Africa? And by what sources or methods can you generalize about it? A critical discussion of your methods and sources is missing in the text.

-        The way in which you paint an image of pre-colonial Africa is strangely ahistorical and I (and Natalie Avalos, I assume) would object to your method of taking ethnic studies as peep into the pre-colonial past (p.10).

Some further issues with the text:

-        Casanova distinguishes the secular as secularism (ideology), secularity (state of things), and secularization (processes). This is from the back of my head, and I might not remember his distinction correctly, but I think that your argument would benefit from dwelling on his distinction a bit more.

-        Your argument on page 2 that religion cannot be defined but we come to know it when we see it (cf. Sharpe) does not help to substantiate your argument. On the same page, you consider religion as practice, feeling, and system of belief. What is ‘religion’ for your argument?

-        ATR in the singular. I agree with your critique of the Western denial to consider ATR as world religion, and I also follow you in the decolonisation as re-naming point. But I disagree with speaking of ATR in the singular (or in the plural for that matter) and I don’t see how or why the codification of African beliefs would help in their decolonization (p.3). ATR is a colonial invention, and it seems to me that your suggestion would be to reclaim and revalue it as a world religion. To me, this leaves open the question what ATR is and how we can subsume Africa’s diverse religious traditions under a single rubric.

-        Birgit Meyer and Olga Linares are both women; their pronouns should be ‘she’ (p.3)

-        Your discussion of the normativity of the concept of ‘religion’ (pp.3-4) is compelling but would benefit from some rephrasing for narrative and readability.

-        Pp.5-7: your discussion of Africa being incurably religious is a bit lengthy and would benefit from you clearly taking sides and making your point.

-        Pp. 7-10: I am completely convinced by your approach to consider secularization as processes of differentiation. The section would benefit from you spelling this out in more detail.

-        Your glances at pre-colonial societies in West Africa are methodologically flawed (see above), quite superficial, and strangely ahistorical. In the space of only four pages, you take four different societies as examples for Africa’s pre-colonial secularism without discussing your sources or properly dating the periods that you write about. Your discussion of historically given secularism in these societies remains shallow and uncompelling. This could be amended if you develop your argument and approach more clearly above and then apply it to these societies as different case studies. Your argument: If we consider secularism as a process of differentiation between the religious and the state, we can ask whether and how such processes occurred in pre-colonial West African societies which we will do in the following by close readings of the histories of pre-colonial Yoruba, Dagomba, Benin, and Jola societies. Thereby, you could substantiate your point and find a narrative to order your text.

-        As an aside: What do you mean by ‘ethnic nationality’ (p.10)?

-        Your notion of ‘religio-secularity’ is introduced only in the Conclusion and remains undefined. What do you mean by this term? And what makes it peculiarly African? I remain yet to understand the notion and to be convinced by it.

-        Rework your language and use of tropes throughout (e.g., ‘The status of phrased religion became the light to see through the heart of the dark continent’, p.4; ‘driven religious consciousness deeper into the psyche of majority of Africans’, p.5; ‘advanced in monarchy’, p.9; ‘The Igbo believed that’, p.10; ‘made it difficult to milk any element of secularism’, p.14).

-        It seems to me that the text was written by different authors. The formatting is inconsistent, the review of the literature reads a bit incoherent, and the different case studies are not jointly discussed. This should be amended.

-        Literature that is central to the text but not discussed in it: Chidester on the colonial invention of religion in Africa, Asad on the Western genealogy of religion and the secular, the extensive discussion of how we can (not) study pre-colonial Africa.

The paper clearly has the potential to become an excellent and much needed contribution to the debate on religion (and secularism) in Africa, and I look forward to reading a revised version of it.

Author Response

I have responded to your comments, and I do appreciate them.

Reviewer 2 Report

This article is somewhere between "accept after minor revision" and "reconsider after major revision" but I have recommended the former because there are good building blocks here, it is mostly a matter of connecting the pieces. The author(s) need to begin by constructing a clear thesis and stating it at the beginning of the work. As it stands, the introduction merely states "This article is therefore poised to examine the trajectories of ‘religion’ and ‘secularism’ in pre-colonial Africa at the turn of colonialism." The author(s) need to answer: "to what end"? What are they trying to say about these trajectories? Why should the reader care?

This is particularly true because the title and introduction give the impression that the author(s)' main focus is actually on 'secularism' not 'religion' so the pages of discussion of 'religion' in Part 2 leave the reader feeling rather lost about what they should be taking away from this part. I strongly recommend a paragraph in the introduction that expands upon the statement at the end of the abstract "Not denying the widespread diversities of religious beliefs in pre-colonial Africa, this paper argues for the presence and praxis of religio-secularity, which is non-atheistic in nature." This seems to be the key argument so they should explain to the reader how each section builds to that point. 

Additionally, the author(s) need to restructure the section on secularism. Like the overall article, the authors should be clearer about where they are going. After reaching the end of the article (and reading the above-mentioned statement in the abstract), it seems that the author(s) are arguing for a distinct interpretation of secularism in Africa. However, this is not clear at all in Part 3. Instead, it jumps from a discussion of the ambiguities of the term in the West to examples of African societies with the author(s) seem to say are just different political ideologies and administrative systems. THe reader needs to know the argument about whether or not there is secularism in these societies (or to what degree there is) before they can understand why these are given as examples. As it stands, the reader has no idea what to take away from these examples until the last sentences of the paper. THen, to evaluate the examples, the reader would have to go back and reread the whole paper. Tell the reader upfront that there could be a different form of secularism and explain why these examples help illustrate that. 

In addition to these points about developing the thesis and the clarity of the argument, it would be helpful to discuss the significance of their argument. The author(s) mention how the achievement of secularism, like the existence of 'religion' is a yardstick for 'civilization.' Thus, it seems very important to address, at least briefly, what happens if we agree that a special kind of secularism exists in Africa. Can this be seen as equal with other forms of secularism or will it be dismissed as a less developed form of secularism (as African religions were often dismissed as less developed forms of religions) that needs to evolve toward a 'purer" form? 

Author Response

Many thanks for your comments. I have responded to them.

Round 2

Reviewer 1 Report

I’ve read the revised article with great interest and think that it has improved. Thank you for considering the reviewers’ suggestions and all the work you’ve put into this. I appreciate it.

This is an interesting paper that raises an important issue, asks a pertinent question, and probes into some noteworthy case studies (problems of method notwithstanding). It should be published after some minor revisions. Given the scale of the paper (pre-colonial Africa writ large and ‘the African worldview’; p.18), I find the argument (religio-secularity as common in pre-colonial African societies) not yet sufficiently substantiated. One way to substantiate the argument would indeed be to cover pre-colonial Africa – a daring and impossible task – and to discuss the methods needed to do so in full detail. Another way to do so would be to rescale the argument. The latter would be my suggestion for the final revisions.

The paper argues compellingly that African religious traditions have long been ill-conceived and discriminated against in Western thought. By the same token, secular Africa has unduly been discounted. Therefrom, the author raises the pertinent question how one could come up with better ways to study and think about religion and secularity in pre-colonial Africa. The main question would be: How can the study of secularity (and religion) in pre-colonial Africa look like? Part of the answer provided by the author is that we reconceptualize religion in Africa (I fully agree). They opt for the ‘incurably religious’ thesis (with which I disagree) but also argue that processes of (social) differentiation were at work in pre-colonial African societies (I agree). They argue that we should consider these processes to study the secular in pre-colonial Africa (I agree). This leads the author to a sweeping assertion that pre-colonial Africa had religio-secularity (I disagree as I find this way too sweeping and generalizing).

To scale down the argument, the paper could keep its excellent discussion of lopsided Western takes on religion and the secular in Africa, raise its pertinent question, and then argue that processes of differentiation were indeed at work in pre-colonial African societies without generalizing about them. The case studies could then be presented as such and not as patchy grounds to make claims about pre-colonial Africa writ large. Highlighting that such processes were indeed present in various pre-colonial African societies (in different forms and dynamics), the author could make the important point that differentiations between the religious and the secular were and are found beyond the West and that an exploration of such processes in pre-colonial Africa offers hitherto neglected grounds to reconceive secularity in Africa (without arguing for a continuity from the pre- to the post-colonial).

Other issues with the text:

p.2: ‘Utilising historical and analytical methods’. Please spell these methods out in more detail. The most important question ‘How can we seize and assess pre-colonial African societies and the (social) concepts of their members?’ remains unanswered (to me). Please elaborate how one can write about secularism in pre-colonial Africa writ large.

p.18 and throughout: How can one write about pre-colonial Africa and ‘the African worldview before the advent of missions and colonialism’ in the singular? This relates to the problems in method and should be commented upon. After all, the paper makes generalizing claims about millions of people over thousands of years. Two caveats: 1. What about Ethiopia? How does the history of Ethiopia fit into your scheme? 2. What about differences between and within pre-colonial African societies?

p.2: ‘African indigenous societies’ or ‘ethnic nationalities’ (p.13) remain ill-defined. Why not just call them ‘different societies’ or ‘different African societies’?

p.2: ‘Religio-secularity thesis is significant for understanding most contemporary African countries where religion still plays critical roles in public policies in spite of the claim that they are practising (constitutional) secularity.’ I find this alleged continuity between pre-colonial and post-colonial thought and life in Africa not yet sufficiently substantiated in the paper. I’d also be interested to learn more about the method by which one can make such claims.

p.3: ‘Religio-secularity differs from secularism in that the latter either aligns with atheism, religious nones, philosophical and ideological disenchantment with religion or binary split between the profane and the sacred, this-worldly and other-worldly.’ Please provide references. This sounds like one reading of the secular among others to me, and as the secularism = atheism part of this sentence seems central to the overall argument, I’d be interested in a more extensive discussion of this point.

p.3: ‘However, one can define religion as the awareness, recognition, consciousness of, and belief in, the existence of a supreme being to whom worship is due.’ This reads like a quite monotheistic and Abrahamic/Western definition of religion to me which the author otherwise criticizes sharply. ‘Religion’ is then defined below as ‘Religion started as soon as a people were conscious of the existence of the supernatural, which they wanted to establish a relationship with.’ (p.5) and juxtaposed to ‘the belief in the presence and efficacy of ancestorship’ (p.18). ‘Religion’ should be used more consistently throughout so that it does not turn into a vague signifier.

p.6: ‘Of all the world religions, only Africa’s will have the word religion as an appellation as though all others are not religions.’ While the author is highly critical of the name given to ATR by Western observers (and rightfully so), they seem to concur with their invention of ‘ATR’. Can we indeed speak of a single and encompassing ATR? Are there many different ATRs which nonetheless have enough in common to be subsumed under the ATR umbrella? I’d like to read a bit more on that.

p.7: ‘The heavy reliance on religion is not just that the people are incurably religious (Idowu 1973) even in post-colonial era as it was in pre-colonial experience (Mbiti 1969), it can be argued that political and socio-economic reality of post-colonial policies have driven religious consciousness deeper into the mind of majority of Africans.’ How so? Here, I read the same strange continuity between pre- and post-colonial Africa as asserted by the author above.

p.13: ‘From Avalos’ (2020) position that area or ethnic studies are instructive to peep into the past’. I don’t think that Avalos would agree with this rendering of her argument (as it is phrased, the reference to her carries the weight of this sentence). Beyond that, I am not convinced that ethnic studies allow us to peep into the past, this sounds like 19th century evolutionary anthropology to me.

p.13: ‘Religio-secularity is the concept that states that social institutions might be differentiated, but they were not entirely independent of religion.’ As this seems to be the core definition of ‘religio-secularity’, I’d like to read a more detailed discussion of the added value of this concept here.

pp.13-17: the case studies. How were these studies selected? How can one generalize about pre-colonial Africa from them? How do the selected studies discuss their methodology of accessing and assessing pre-colonial African societies?

p.16: ‘The four cases examined above demonstrated that despite the claim of African religious incurability, the societies were organised politically to check the excesses of both religious and political institutions.’ This seems to me to be the core finding of this paper: Processes of differentiation were present in different forms and dynamics in various pre-colonial African societies. This makes for a highly important finding that offers compelling insights into religion-secular beyond the West.

Author Response

Kindly find attached the explanations to the queries.
